# Quasi-two-dimensional superconductivity from dimerization of atomically ordered AuTe$_2$Se$_{4/3}$ cubes

J.G. Guo [1], X. Chen [1,2], X.Y. Jia[3], Q.H. Zhang[1], N. Liu[1,2], H.C. Lei[4], S.Y. Li[3,5], L. Gu[1,6,7], S.F. Jin[1,6] & X.L. Chen[1,6,7]

The emergent phenomena such as superconductivity and topological phase transitions can be observed in strict two-dimensional (2D) crystalline matters. Artificial interfaces and one atomic thickness layers are typical 2D materials of this kind. Although having 2D characters, most bulky layered compounds, however, do not possess these striking properties. Here, we report quasi-2D superconductivity in bulky AuTe$_2$Se$_{4/3}$, where the reduction in dimensionality is achieved through inducing the elongated covalent Te–Te bonds. The atomic-resolution images reveal that the Au, Te, and Se are atomically ordered in a cube, among which are Te–Te bonds of 3.18 and 3.28 Å. The superconductivity at 2.85 K is discovered, which is unraveled to be the quasi-2D nature owing to the Berezinsky–Kosterlitz–Thouless topological transition. The nesting of nearly parallel Fermi sheets could give rise to strong electron–phonon coupling. It is proposed that further depleting the thickness could result in more topologically-related phenomena.

[1] Beijing National Laboratory for Condensed Matter Physics, Institute of Physics, Chinese Academy of Sciences, P.O. Box 603, Beijing 100190, China. [2] University of Chinese Academy of Sciences, Beijing 100049, China. [3] State Key Laboratory of Surface Physics, Department of Physics, and Laboratory of Advanced Materials, Fudan University, Shanghai 200433, China. [4] Department of Physics and Beijing Key Laboratory of Opto-electronic Functional Materials and Micro-nano Devices, Renmin University of China, Beijing 100872, China. [5] Collaborative Innovation Center of Advanced Microstructures, Nanjing 210093, China. [6] School of Physical Sciences, University of Chinese Academy of Sciences, Beijing 100049, China. [7] Collaborative Innovation Center of Quantum Matter, Beijing 100084, China. Correspondence and requests for materials should be addressed to J.G.G. (email: jgguo@iphy.ac.cn) or to X.L.C. (email: chenx29@iphy.ac.cn)

The dimensional reduction or degeneracy usually induces the significant change of electronic structure and unexpected properties. The monolayer, interface and a few layers of bulky compounds are typical resultant forms of low dimensionality. The two dimensional (2D) material, for instance, graphene, is found to have a linear energy dispersion near Fermi energy ($E_F$) and possess a number of novel properties[1–3]. Monolayer $MoS_2$ exhibits a direct energy gap of 1.8 eV[4] and pronounced photoluminescence[5], in contrast to trivial photoresponse in bulky $MoS_2$ with an indirect band-gap.

2D superconductivity (SC), a property closely related to dimensional reduction, has been observed in a variety of crystalline materials like $ZrNCl$[6], $NbSe_2$[7], and $MoS_2$[8] recently through the electric-double layer transistor (EDLT)[9] technique. Many emerged properties, i.e., the well-defined superconducting dome, metallic ground state and high upper critical field[6, 8], significantly differ from those of intercalated counterparts. Besides, the lack of in-plane inversion symmetry in the outmost layer of $MoS_2/NbSe_2$ with strong Ising spin-orbital coupling induces a valley polarization[7, 8]. In the scenario of low-dimensional interface, the unexpected 2D SC[10, 11], the remarkable domed-shaped superconducting critical temperature ($T_c$)[12], pseudo-gap state[13], and quantum criticality[14] have been demonstrated in La(Al,Ti)$O_3/SrTiO_3$(001) film. Very recently, the interface between $Bi_2Te_3$ and FeTe thin films displayed 2D SC evidenced by Berezinsky–Kosterlitz–Thouless (BKT) transition at 10.1 K[15]. The tentative explanations are related to the strong Rashba-type spin–orbit interactions in the 2D limit.

At the moment, the way to fabricating low-dimensional materials generally involves molecule beam epitaxy and exfoliation from the layered compounds. The top-down reduction processes usually are sophisticated and time consuming for realizing scalable and controllable crystalline samples. There are other chemical routes to tuning dimensionality by means of either changing the size of intercalated cations or incorporating additional anions. It is reported that increasing the size of alkaline-earth metals Ae (Ae=Mg, Ca, and Ba) between [NiGe] ribbons can reduce three dimensional (3D) structure to quasi-1 dimensional one[16]. In addition, the ternary CaNiGe can be converted into ZrCuSiAs-type CaNiGeH by forming additional Ca–H bonds, which exhibits different properties owing to the emergence of 2D electronic states[17]. The metastable $Au_{1−x}Te_x$ ($0.6 < x < 0.85$) show an $\alpha$-type polonium structure[18, 19], in which the Au and Te disorderly locate at the eight corners of a simple cubic unit cell. The $T_c$ fluctuates in the range of 1.5–3.0 K, but the mechanism of SC has been barely understood[20]. Besides, the equilibrium phase $AuTe_2$, known as calaverite, is a non-superconducting compound, in which distorted $AuTe_6$ octahedra are connected by Te–Te dimers[21].

Through incorporating more electronegative Se anions, we fabricate a new layered compound $AuTe_2Se_{4/3}$ by conventional high temperature solid-state reaction. In a basic cube subunit, the Se anions attract electrons from Te and lead to the ordered arrangement of Au, Te and Se atoms. The cubes stack into strip through Te–Te dimers at 3.18 Å and 3.28 Å along the $a$- and $b$-axis, respectively, which composes 2D layers due to the existence of weak Te–Te interaction (~4 Å) along the $c$-axis. Electrical and magnetic measurements demonstrate that the SC occurs at 2.85 K. Furthermore, this SC exhibits 2D nature evidenced by the BKT transition in the thin crystals. The observed results are interpreted according to the crystallographic and electronic structure in reduced-dimensionality.

## Results

### Structural characterization
Figure 1a shows the scanning electron microscope (SEM) image of $AuTe_2Se_{4/3}$ crystal, and the typical shape is like a thin rectangle. Its anisotropic morphology implies that there exist weak bonds and the weakest one determines the most cleavable face in $AuTe_2Se_{4/3}$. The elemental mapping shows homogeneous distribution of Au, Te, and Se atoms. The atomic ratio is Au:Te:Se = 25:46:29, as EDS pattern in Fig. 1b. Most reflections except a few weak peaks in the experimental powder X-ray diffraction (PXRD) pattern (Supplementary Fig. 1a) of as-grown sample can be indexed based a triclinic unit cell. Efforts for determining the crystal structure by solving PXRD pattern have failed.

We exfoliated the crystals by Scotch tape and obtained thin enough samples for observing the HAADF images. As the contrast of HAADF image exhibits a $Z^{1.7}$ dependent relation, where $Z$ is the atomic number, different kinds of atom columns can be directly distinguished (i.e., in HAADF images, the largest contrast indicates Au columns). Figure 1e delineates the atomic arrangement of the $ab$-plane of $AuTe_2Se_{4/3}$. It can be seen that elongated atoms locate equispaced at 2.69 Å along two orthogonal directions based on the peak interval in Fig. 1c. The corresponding selected area electron diffraction (SAED) pattern along [1–39] zone axis in Fig. 1d confirms the tetragonality of the $ab$-plane. It is noted that two kinds of superstructure spots are present, one being along $a^*$ with a new $a^*/3$ periodicity corresponding to a triple in real space and other plane spacing is $a^*/3 + b^*$.

After tilting the $ab$-plane by ~13° along diagonal Kikuchi line (Supplementary Fig. 2), clear planar 2×2 blocks appear as the image in Fig. 1h. Closer examination of a block reveals that two separated atoms exist at the center spots and hence elongated feature shows up, which actually comes from the tilted top and bottom atoms in a block (Supplementary Fig. 3). Each block is separated by 3.18 Å and 3.20 Å along transversal and longitudinal directions as indicated by the red and blue dash lines in Fig. 1f and i, respectively. The bond lengths are periodically modulated by two short bonds and one long bond (···2.69 Å–2.69 Å–3.18 Å···) and (···2.69 Å–2.69 Å–3.20 Å···), respectively. Figure 1g, j and m show the SAED patterns along the [00–1], [−10–3], and [−100] zone axis. Again, one can see that a modulation of $a^*/3$ exists. The HAADF images of Fig. 1k and i present that the bright spots are sandwiched by two weak spots separated by 2.69 and 3.20 Å, indicating the atomic distributions of front/back face differ from those of middle layer in each block. To observe how the blocks stack along the weakly combined direction, namely the $c$-axis, we prepare narrow samples (~50 nm) by focused ion beam (FIB) milling. The HAADF image in Fig. 1n reveals that the building blocks are 3.20 Å and 3.47 Å apart along two directions, as shown in Fig.1l, respectively. So the periodic bond lengths along $c$-axis can be described as ···2.69 Å-2.69 Å-3.47 Å···.

On the basis of above observations, the crystal structure can be regarded as the stacking of cube subunits in real space. The stereotype picture of a block is drawn in Fig. 2a. Inside the block, all of the atoms are orderly arranged and connected by bonds with identical length 2.69 Å. Each Au atom is four planar-coordinated by Te, and each Se atom is connected by three Te atoms. The body center is vacant, which could account for the elongated spots in Fig. 1d. The atomic distribution of front/back face of the cube is $AuTe_4Se_4$, while the middle layer is $Au_4Te_4$. This ordered structure is intimately related to the disorder phase $Au_{1−x}Te_x$ ($0.6 < x < 0.85$) with simple cubic structure. The arrays of cubes in the $ab$-plane are drawn in Fig. 2b. There are one Te–Te dimer (3.18 Å) and three Te–Te dimers (two at 3.18 Å and one at 3.28 Å) in the front/back face and middle layer, respectively. Figure 2c and e shows the front/back and middle strips that are constructed by two kinds of layers. The corresponding electron density difference (EDD) slices from DFT calculations are shown in Fig. 2d and 2f, respectively. One can see that although covalent bonds dominate the whole cube,

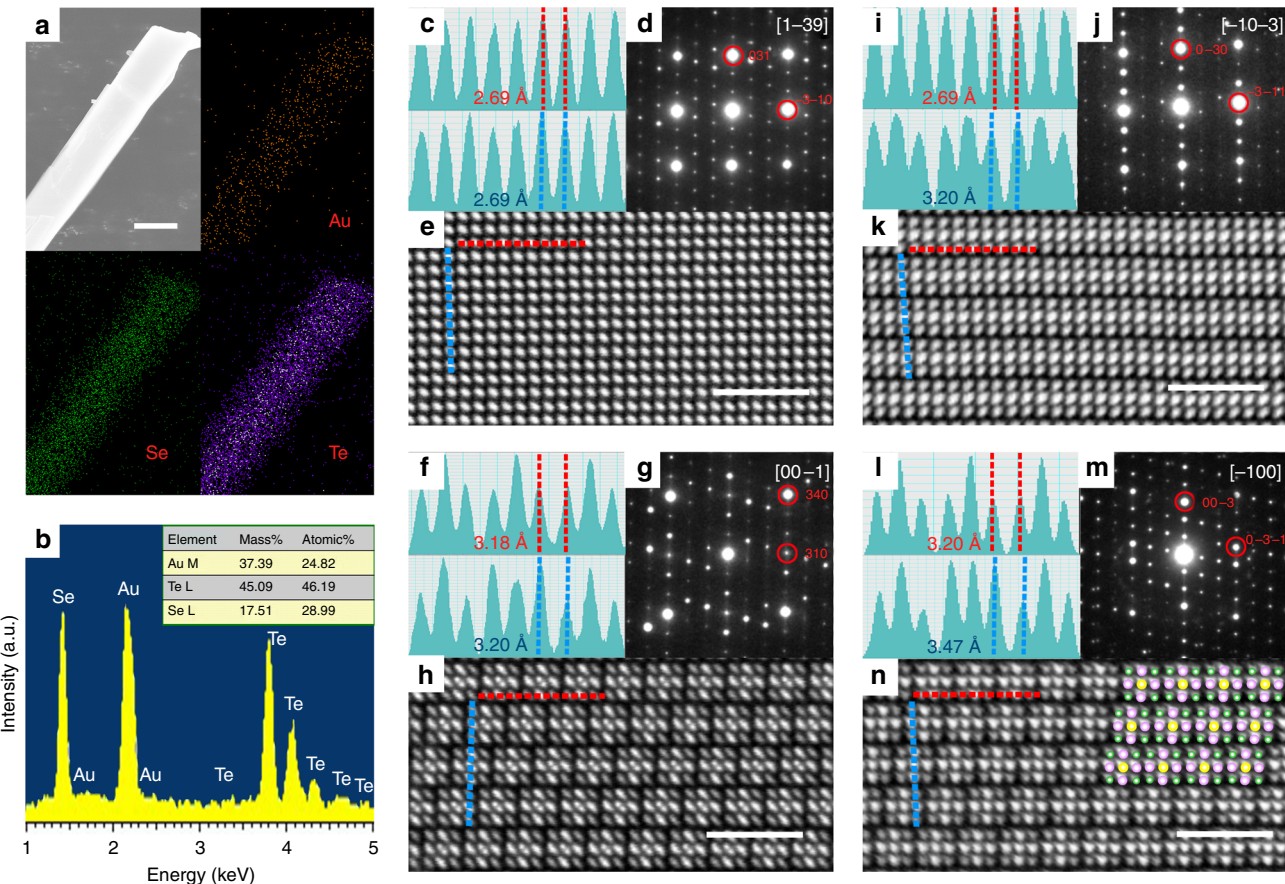

**Fig. 1** Compositional mapping and HAADF images of AuTe$_2$Se$_{4/3}$. **a** The SEM image of AuTe$_2$Se$_{4/3}$ and the EDS mapping images of Au (golden), Se (green), and Te (violet) in AuTe$_2$Se$_{4/3}$. The scale bar here represents 2 μm. **b** The elemental ratio of Au, Te, and Se obtained from the EDS mapping. The two kinds of atomic distances (red and blue color) are estimated along **c** [1–39], **f** [00–1], **i** [−10–3], and **l** [−100] zone axes, respectively. The SAED images are taken along **d** [1–39], **g** [00–1], **j** [−10–3], and **m** [−100] zone axes, respectively. The HAADF images show atomic distributions of Au, Te, and Se along **e** [1–39], **h** [00–1], **k** [−10–3], and **n** [−100] zone axes in real space, respectively. The scale bars here represent 2 nm. The Au, Te, and Se atoms are superimposed on the spots in **n**, as the structural analyses (see the text)

the Au atoms loss electrons in Au–Te bonds and Se atoms attract electrons to some extent. It is noted that the Te–Te bonds of middle layer accumulate more electrons than that of front/back face. These covalent Te–Te bonds among cubes suggest that the electrons can mainly populate in the middle layers along the $a$-axis and are perturbed by the coordinating Se atoms. Besides, the longer Te–Te bond lengths (3.28 Å) link horizontal-shifted strips, forming 2D stacking of cubes in the $ab$-plane.

The determined crystal structure of AuTe$_2$Se$_{4/3}$ is shown in Fig. 2g (see detail crystallographic parameters in Supplementary Table 1). It possesses a triclinic unit cell with $P$-1 (No. 2) symmetry and the lattice parameters are $a = 8.85$ Å, $b = 8.43$ Å, $c = 9.28$ Å, $\alpha = 77.24°$, $\beta = 95.53°$, and $\gamma = 72.36°$. The atomic ratio is Au:Te:Se = 3:6:4 deduced from crystallographic symmetry, which shows a little discrepancy from the measured value owing to the uncertainty in the EDS method. One can see that the spacings among cubes are 3.18 Å, 3.20 Å and 3.47 Å along the $a$-, $b$- and $c$- axis, respectively. The longest Te···Te distance between the $ab$-plane is ~ 4 Å, which is larger than the Te–Te bond length in element Te[22], the Van der Waals bonds of MoS$_2$ (3.48 Å)[23] and graphite (3.37 Å)[1]. The simulated PXRD pattern based on the present crystallographic parameters is plotted in Supplementary Fig. 1b. All the indices of high-intensity peaks of experimental PXRD can match the simulated ones very well except minor impurity peaks. Furthermore, the atomic arrangements viewed from [1–39], [00–1], [−10–3], and [−100] zone axes are highly

consistent with experimental HAADF images (Supplementary Fig. 3), demonstrating the validation of present structure within the accuracy of electron microscopy.

Figure 3a presents the XPS pattern of Au 4$f$, Te 3$d$, and Se 3$d$ core-level measured at 300 K. The binding energy of Te 3$d_{5/2}$, 573.2 eV, is close to the value in element Te (573.0 eV), indicating the 5$p$ orbitals of Te are not fully occupied and responsible for the conducting electrons. The binding energy of Au 4$f_{7/2}$ is 84.6 eV, which is close to 84.4 eV (Au$^+$) and smaller than 85.2 eV (Au$^{3+}$)[24]. For the Se 3$d_{3/2}$, the binding energy is 54.2 eV, which is close to the value in CdSe (54.4 eV) and obviously smaller than that in Se element (55.1 eV). Therefore, the charge balanced formula of AuTe$_2$Se$_{4/3}$ can be written as $[\mathrm{Au}^{(1+\delta)+}][(\mathrm{Te}_2)^{(5/3-\delta)+}][(\mathrm{Se}^{2-})_{4/3}]$. Figure 3b shows the band structure of AuTe$_2$Se$_{4/3}$. The flat bands lead to the multiple Van Hove singularities in the plots of density of states. It can be seen that there are four bands, denoted as I, II, III, and IV bands, crossing the $E_\mathrm{F}$. The I, III, and IV bands cross the $E_\mathrm{F}$ along G-F, G-B, and Z-G in first Brillouin zone. The II band only crosses the $E_\mathrm{F}$ along G-F, which exhibits a nearly flat dispersion along F-Q with quasi-1D character. Figure 3c shows the projected density of states (PDOS) around the $E_\mathrm{F}$. The total DOS at the $E_\mathrm{F}$ is estimated to be 6.5 states eV$^{-1}$ per unit cell, and the contribution of Te 5$p$ states is ~70%, which are the primary source of conductive electrons. The whole Fermi surface of AuTe$_2$Se$_{4/3}$ and four separated sheets are individually plotted in Fig. 3d. One can find that the Fermi surface consists of three 3D

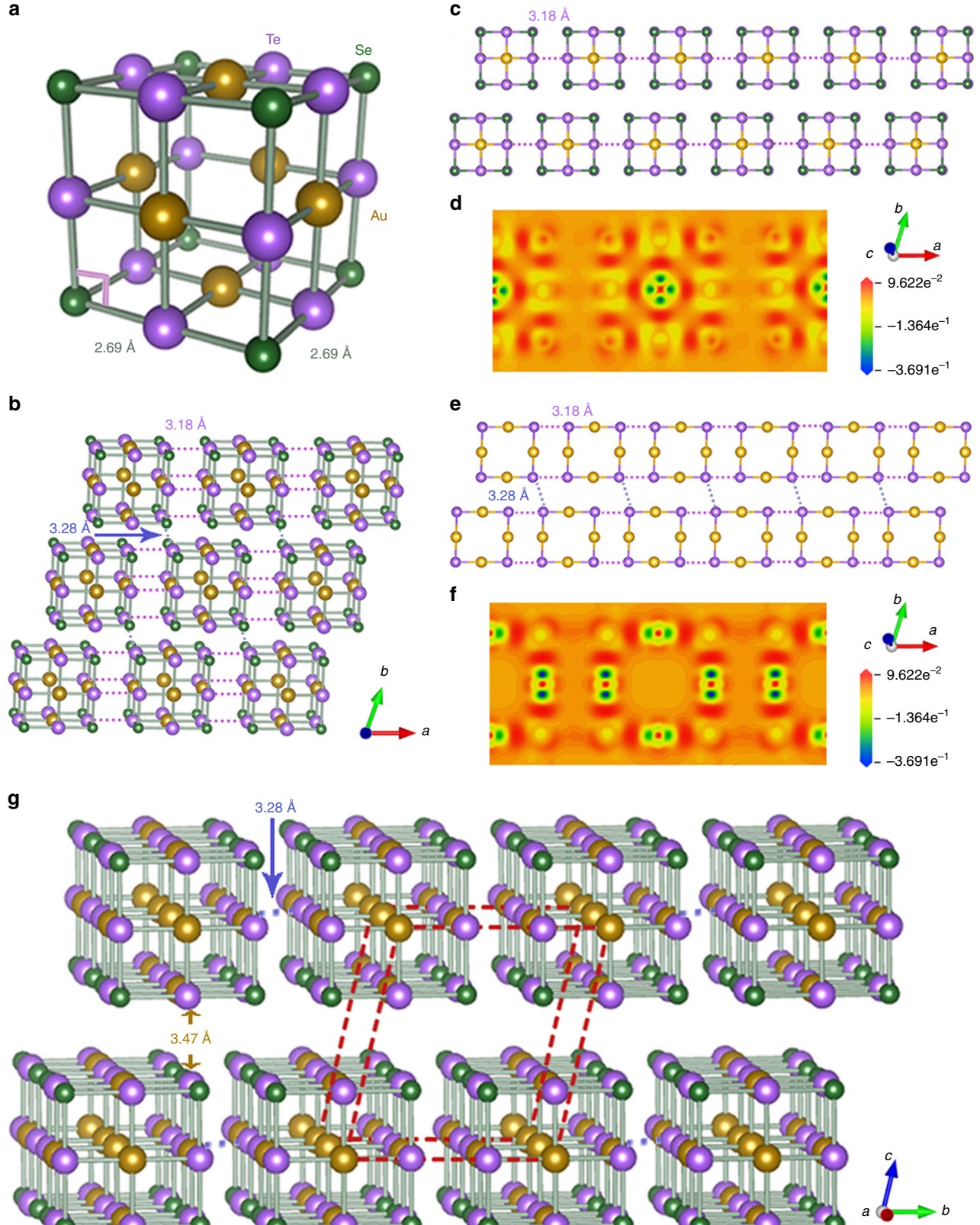

**Fig. 2** Crystal structure and electron density difference of AuTe$_2$Se$_{4/3}$. **a** The structure of single block of atomically ordered Au–Te–Se. **b** The stereotype structure of the *ab*-plane with two kinds of Te–Te bond lengths of 3.18 and 3.28 Å. **c** The strips of front/back face of cubic arrays. The violet dash lines denote the Te–Te bonds of neighbor cubes along the *a*-axis. **d** The calculated EDD of front/back face, which is calculated by subtracting free atom electron density from the total electron density. The plus and minus scale represent the accumulated and depleted electrons. **e** The strips of middle layer of cubic arrays. The two violet and one blue dash lines denote the Te–Te bonds of neighbor cubes. **f** The corresponding EDD images of middle layer. **g** The whole crystal structure of AuTe$_2$Se$_{4/3}$. The dash lines represent the triclinic unit cell ($P$–1)

sheets from the I, III, and IV bands and one quasi-1D Fermi sheets from the II band. The last sheets are nearly perpendicular to the *a*-axis in real space, which means that the conductive electrons mainly flow along this direction. In addition, it should be emphasized that the strong nesting between quasi-parallel Fermi sheets will enhance the electron–phonon coupling, which possibly favors the SC.

**Superconductivity**. Figure 4 shows the electrical transport and magnetic properties for $AuTe_2Se_{4/3}$. The electrical resistivity exhibits metallic conductivity, as the plot in Fig. 4a, which is consistent with the finite DOS value at the $E_F$. The residual resistivity ratio (RRR) is ~20, suggesting the good quality of sample. The resistivity in low-temperature range can be well fitted using $\rho_{(T)} = \rho_{(0)} + AT^2$ equation, which indicates the sample is a typical Fermi-liquid system. The superconducting transition occurs at $T_c^{onset} = 2.85$ K. The drop of resistivity can be described by Aslamazov–Larkin[25] and Maki–Thompson[26, 27] formulas that account for superconducting fluctuations. Our fitting results indicate that the finite scale of 2D order parameters emerge below $T_{c0} = 2.81$ K[28, 29], see Supplementary Fig. 4. We find that temperature dependent upper critical fields obey 2D SC characters below $T_c$. As seen from Fig. 4b and 4c, $T_c$ monotonically decreases under in-plane ($H//ab$) and out-of-plane ($H//c$) external magnetic fields. The two $\mu_0H_{c2}(0)$, as shown in Supplementary Fig. 5a, can be determined by the Ginzburg–Landau (GL) expressions for 2D SC

$$\mu_0 H_{c2}^{//c} = \frac{\Phi_0}{2\pi\xi_{GL}^2(0)}\left(1 - T/T_c\right) \tag{1}$$

$$\mu_0 H_{c2}^{//ab} = \frac{\Phi_0\sqrt{12}}{2\pi\xi_{GL}(0)d_{sc}}\sqrt{1 - T/T_c} \tag{2}$$

where $\Phi_0$ is the flux quantum, $\xi_{GL}(0)$ the GL coherence length at T=0 K, and $d_{sc}$ the SC thickness. Fitting the data of $\mu_0H_{c2}^{//c}$ against temperature yields $\xi_{GL}(0) = 28.7$ nm. From the second equation, we can extract the value of $d_{sc}$ is 10.8 nm, which is smaller than $\xi_{GL}(0)$, demonstrating the SC is quasi-2D. One thing should be noted is that the relatively large $d_{sc}$ might originate from small misalignment since the $AuTe_2Se_{4/3}$ thin flakes is easily-bent. To confirm the 2D SC, we measured the variation of field-dependent resistivity with different angles at 2.0 K, see Fig. 4d. The schematic diagram of the measurement is shown in the lower inset of Fig. 4e. The extracted angular dependent upper critical fields at 10% of normal resistivity are plotted in Fig. 4e. A clear cusp is observed at $\theta=90°$, at which the magnetic field parallels to the *ab*-plane. The overall $H_{c2}(\theta)$ data can be fitted by Tinkham formula for 2D SC[30]

$$\left|\frac{H_{c2}(\theta)\sin\theta}{H_{c2}^{//c}}\right| + \left(\frac{H_{c2}(\theta)\cos\theta}{H_{c2}^{//ab}}\right)^2 = 1$$

In comparison, the experimental data clearly deviates the curve of 3D anisotropic mass model $H_{c2}(\theta) = H_{c2}^{//ab}/(\sin^2\theta + \gamma^2\cos^2\theta)^{1/2}$; $\gamma = H_{c2}^{//ab}/H_{c2}^{//c}$ around $\theta = 90°$, as shown in the upper inset of Fig. 4e, unambiguously showing the SC of $AuTe_2Se_{4/3}$ is quasi-2D nature.

The SC of $AuTe_2Se_{4/3}$ was examined by the magnetic susceptibility ($\chi$) in the low-temperature range. Figure 4f plots the temperature dependent $\chi$ of $AuTe_2Se_{4/3}$ in zero-filed cooling (ZFC) and field-cooling (FC) mode under 10 Oe. The diamagnetic signals are observed below 2.78 K. The superconducting volume fraction is estimated to be 90% at 1.8 K. The different magnitudes of $\chi$ under $H//ab$ and $H//c$ modes indicate a strong anisotropy of $AuTe_2Se_{4/3}$ single crystal. The magnetization curves

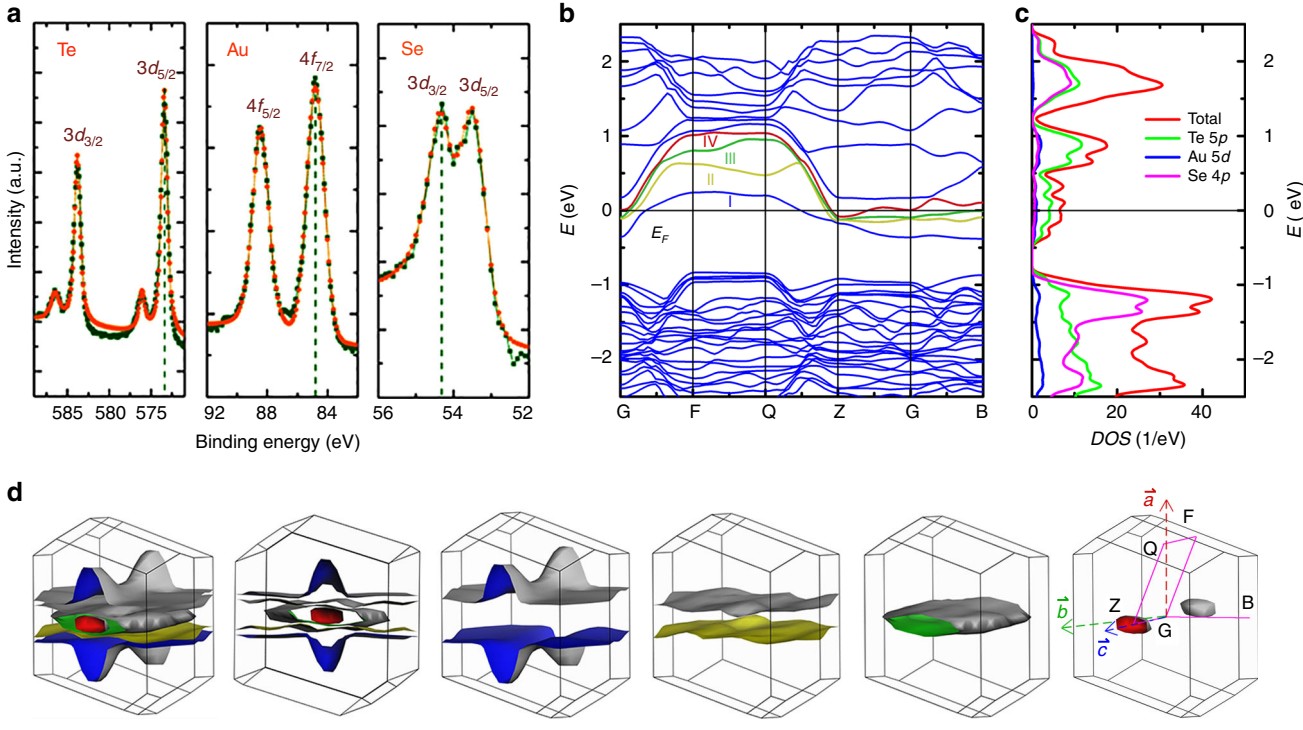

**Fig. 3** Electronic structure of $AuTe_2Se_{4/3}$. **a** The XPS pattern of $AuTe_2Se_{4/3}$ at 300 K. The olive and orange dots and lines are the measured data and fitted curves. **b** The scalar relativistic band structure of $AuTe_2Se_{4/3}$. The four bands crossing the Fermi level are colored as blue, yellow, green, and red. **c** The projected density of states of $AuTe_2Se_{4/3}$. **d** The Fermi sheets of $AuTe_2Se_{4/3}$ at 3D view, side view and four separated bands in the first Brillouin zone. The G point is the zone center. The coordination *a*, *b*, and *c* are the real space notions

around $T_c$ are shown in Fig. 4g, in which the type-II SC is observed. One can obtain the lower critical field $\mu_0 H_{c1}(0)$, 8.2 mT, from fitted curve of the minima of all the $M$–$H$ curves in Supplementary Fig. 5b. The penetration depth $\lambda(0)$ is estimated as 283.5 nm, which is comparable to the value in tellurides[31].

**2D BKT transition**. It is known that the BKT transition can be detected from the slope evolutions of standard $I$–$V$ curves around $T_c$ in 2D superconductors. Figure 5a plots the $I$–$V$ curves in log–log scale near $T_c$, in which the critical $I_c$ increases as the temperature decreases and finally reaches 5 mA at 1.6 K. The $I$–$V$ curve at 3.0 K shows a typical ohmic conductivity. The exponent, $\eta$, from power law $V \sim I^\eta$ increases from 1 and then rapidly as temperature approaching $T_c$ from the high temperature side in Fig. 5b. The $\eta$ reaches 3 at $T_{BKT} = 2.78$ K, which is the signature of BKT transition. One thing should be noted that the fitting should be done at the low-current limit[32]. Moreover, in a narrow temperature range just below $T_c$, it is theoretically proposed that the resistance would follow the equation $R = R_0 \exp[-b/(T-T_{BKT})^{1/2}]$,

where $R_0$ and $b$ are material dependent parameters. Figure 5c plots $(\mathrm{d}\ln(R)/\mathrm{d}T)^{-2/3}$ against temperature from 2.80 to 2.68 K. $T_{BKT}$ is extrapolated to be 2.69 K. The self-consistent $T_{BKT}$ could further confirm the 2D SC nature in AuTe$_2$Se$_{4/3}$. The BKT transition is believed to be related to topological elementary excitations, in which planar vortices and anti-vortices are starting to bound together without the symmetry breaking of order parameters below the $T_{BKT}$[33, 34].

## Discussion

It is known that each Te atom is coordinated by six Te atoms with Te–Te bonds of 2.86 Å (two) and 3.47 Å (four), which make up a series of highly distorted octahedra in hexagonal Te[35]. Stretching the Te–Te bonds and reducing the distortion of octahedral could change the hexagonal ($P3_121$) structure into simple cubic one ($Pm$-3m) through elemental substitutions[22]. As previous reports[18, 36], the Au substitution for Te could average the bond lengths and form a regular octahedron in Au$_{1-x}$Te$_x$ ($0.6 < x < 0.85$) with simple cubic structure. This, however, is not

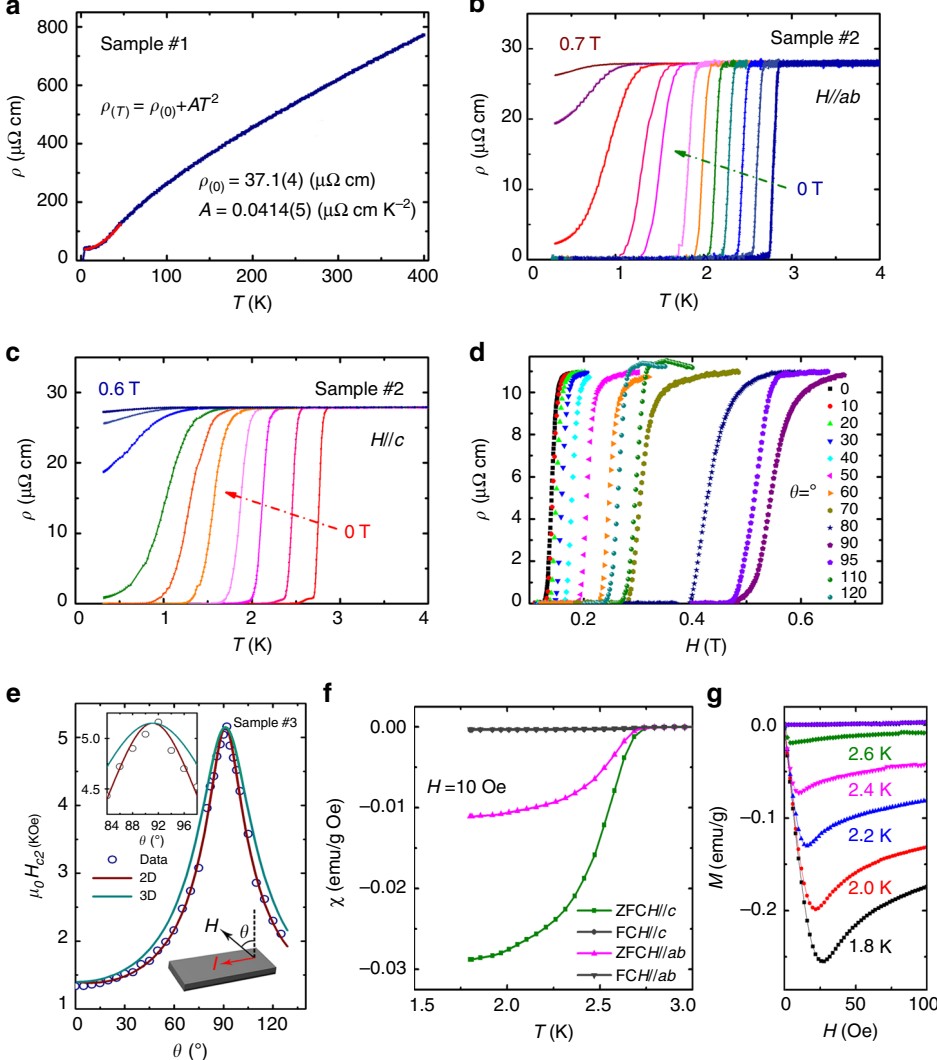

**Fig. 4** Transport and magnetic properties of AuTe$_2$Se$_{4/3}$. **a** The electrical resistivity of AuTe$_2$Se$_{4/3}$ from 400 to 2 K. The quadratic fitting in the low temperature range exhibits the Fermi-liquid behavior. **b**, **c** Superconducting transition of AuTe$_2$Se$_{4/3}$ along $H//ab$ and $H//c$ under magnetic field. **d** The magnetic field dependent resistivity of different angles ($\theta$) at 2 K. **e** Angular dependence of the upper critical field $\mu_0 H_{c2}(\theta)$. The upper inset shows the magnified view of the range near $\theta = 90°$. The $\theta$ is the angle between magnetic field and the normal direction of the $ab$-plane. The lower inset shows the schematic diagram of measurement assembly. The wine and cyan solid lines are the fitted curves with 2D and 3D model as discussed in the text. **f** Temperature dependence of magnetic susceptibility ($\chi$) with ZFC and FC mode. **g** The magnetization curves of AuTe$_2$Se$_{4/3}$ at different temperature

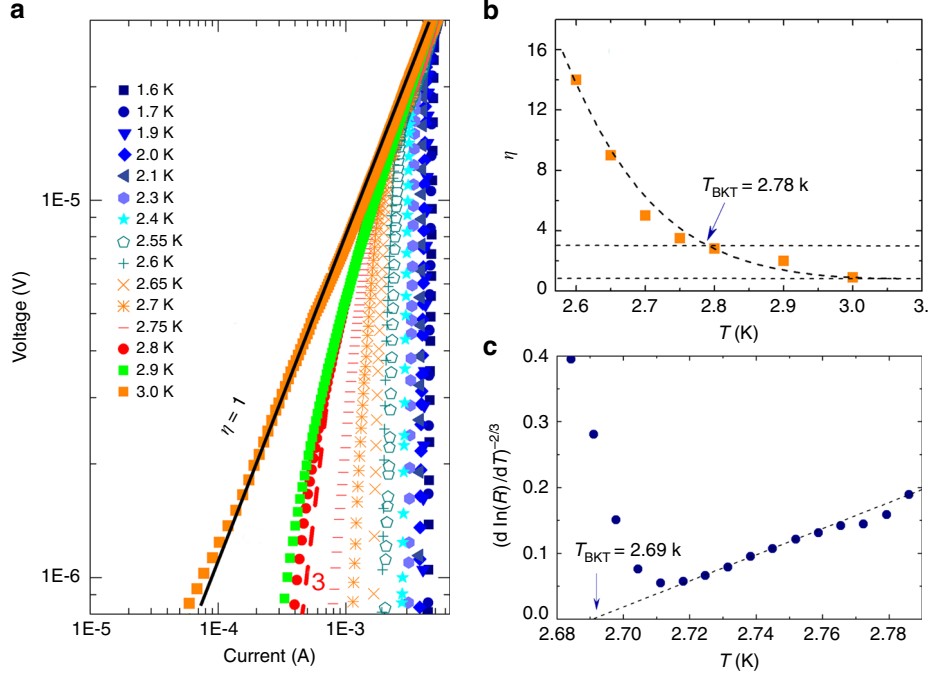

**Fig. 5** Quasi-two-dimensional nature of superconductivity in $AuTe_2Se_{4/3}$. **a** $I$–$V$ curves plotted in a log–log scale at various temperatures near $T_c$. Dash lines represent the $\eta = 1$ (black) and 3 (red) curves, respectively. **b** Temperature dependence of the exponent $\eta$ deduced from the power-law fitting. Two straight dash lines represent the $\eta = 1$ and 3. **c** $R(T)$ curves plotted as $[d\ln(R)/dT]^{2/3}$ versus $T$. The dash line extrapolates the expected BKT transition at $T_{BKT} = 2.69$ K

energetically favorable as the cubic structure only is obtained by the quenching method. Hence, it is reasonable to infer that the total energy of the cubic phase might decrease if Au and Te atoms are ordering. Introduction of more electronegative Se atoms is thought to enhance the ordering according to our results. Meanwhile, as the bond strength of Au–Te and Se–Te bonds are stronger than that of Te–Te bond, the horizontal shift of cubes could increase the amount of Au–Te and Se–Te bonds and further consume electrons in the $ab$-plane. Therefore, the $ab$-plane is nearly electrical neutrality and the interaction of interlayer is Van der Waals force. On the other hand, inside the $ab$-plane, the shortest Te–Te bond (3.18 Å) is thought to be the origin of low-dimensional electronic structure, and another Te–Te bond (3.28 Å) might be the origin of the 2D morphology of $AuTe_2Se_{4/3}$.

In equilibrium phase $AuTe_2$, it was reported that structural transition from monoclinic to trigonal symmetry was accompanied by the appearance of SC, which can be induced by substitution or application of physical pressure[37, 38]. The Te–Te dimers are claimed to compete with SC, where the breaking of Te–Te bonds could enhance the DOS and hence SC ensues. The Te $5p$ states in $AuTe_2$ are responsible for the conductivity[39], which are similar to those of $Au_{1-x}Te_x$ and $AuTe_2Se_{4/3}$. As the DFT calculations, the Fermi surface of $Au_{1-x}Te_x$ displays 3D type (Supplementary Fig. 6). The incorporating Se atom in $Au_{1-x}Te_x$ lowers the structural dimensionality and electronic structure to 2D, which causes approximately parallel Fermi sheets of $AuTe_2Se_{4/3}$. These peculiar Fermi sheets usually involve charge/spin–density–wave instability or non-Fermi liquid behavior in Bechgaard salts $(TMTSF)_2PF_6$[40] and $K_2Cr_3As_3$[41] superconductors, in which the linear resistivity and large $\mu_0H_{c2}(0)$ are beyond the conventional BCS theory[42]. However, the observations of Fermi-liquid type of normal-state resistivity and a small value of $\mu_0H_{c2}(0)$ imply that $AuTe_2Se_{4/3}$ is likely to be a conventional superconductor.

To the best of our knowledge, the $AuTe_2Se_{4/3}$ is a unique layered compound that is realized by anisotropic linking of cubes

with diverse Te–Te bond lengths. The reduced-dimensionality by introducing Se atoms into $AuTe_2$ is significant for the emergence of quasi-2D SC. Recent experiments suggest that the reducing thickness of superconducting materials can facilitate 2D SC in ultra-thin $NbSe_2$[43, 44], $FeSe$[45], and $Ga$[46] films. Simultaneously, a series of profound properties like coexistence of SC and charge–density–wave, enhanced $T_c$ and quantum Griffiths phase are accompanied as well. The $AuTe_2Se_{4/3}$ is easily cleaved to thin crystals, offering a new playground for studying surface or edge states in the $ab$-plane. Furthermore, the topological transition of SC is demonstrated in $AuTe_2Se_{4/3}$, which would be more significant for examining the topological SC in monolayer limit. At the same time, the $T_c$ and $\mu_0H_{c2}(0)$ can be remarkably enhanced through spin–orbital coupling, and a series of quantum phenomena are highly expected due to the large Rashba-type coupling[12, 47] in high-$Z$ elements Au and Te.

## Methods

**Synthesis**. Single crystals of $AuTe_2Se_{4/3}$ were grown using the self-flux method. Total 2–4 g starting materials with high purity Au powder (99.99%, Sigma Aldrich), Te powder (99.999%, Sigma Aldrich), and Se powder (99.99%, Sigma Aldrich) were stoichiometrically weighted and sealed in an evacuated silica tube in high vacuum ($10^{-5}$ mbar) and subsequently mounted into a muffle furnace. The furnace was heated up to 800 °C in 40 h and dwelled 10 h. Afterward, the furnace was slowly cooled down to 450 °C in 4 days and then shut down. Separated crystals from ingot are generally thin with maximum planar size $4 \times 1$ mm². The single crystals of $AuTe_2Se_{4/3}$ were ribbon shape with shining mirror-like surfaces.

**Characterization**. The powder X-ray diffraction (PXRD) pattern of polycrystalline $AuTe_2Se_{4/3}$ were measured by Panalytical X'pert diffractometer with Cu-K$_\alpha$ anode ($\lambda = 1.5408$ Å). The scanning electron microscopy (SEM) image of single crystal was captured from Hitachi S-4800 FE-SEM. The element mapping and composition of the sample were determined by Energy Dispersive Spectroscopy (EDS). The real composition was averaged as 10 sets of data. The high-angle annular-dark-field (HAADF) images were obtained using an ARM-200F (JEOL, Tokyo, Japan) scanning transmission electron microscope (STEM) operated at 200 kV with a CEOS Cs corrector (CEOS GmbH, Heidelberg, Germany) to cope with the probe-forming objective spherical aberration. The attainable resolution of the probe defined by the objective pre-field is 78 picometers. The focused ion beam (FIB)

method was used to cut narrow sample (width: ~50 nm and thickness 20–50 µm) for structural analysis. The valences of Au, Te, and Se were examined by X-ray photoelectron spectroscopy (XPS) using a JEOL JPS9200 analyzer. Samples were cut into long stripes and followed by attaching four silver wires with silver paint. For the resistance measurement, chromium (5 nm) and silver electrodes (150 nm) were deposited first on the freshly cleaved samples by using thermal evaporation through stencil mask. The transport measurements were carried out by using ac lock-in method and performed down to 300 mK in an Oxford Cryostat equipped with Helium-3 and Helium-4 insert. The dc magnetic properties were characterized using a vibrating sample magnetometer (MPMS-3, Quantum Design).

**DFT calculation**. All the first-principles calculations were performed in the Cambridge Serial Total Energy Package with the plane-wave pseudopotential method. We adopted generalized gradient approximation in the form of Perdew–Burke–Ernzerhof function for exchange-correlation potential. The self-consistent field method was used with a tolerance of $5.0 \times 10^{-7}$ eV atom$^{-1}$. We used ultrasoft pseudo-potentials with a plane-wave energy cutoff of 360 eV. The first Brillouin zone is sampled with grid spacing of 0.04 Å$^{-1}$. Fully optimization of the atomic positions and lattice parameters of compounds were carried out by the Broyden–Fletcher–Goldfarb–Shannon (BFGS) method until the remanent Hellmann–Feynman forces on all components are smaller than 0.01 eV Å$^{-1}$.

**Data availability**. The data that support the findings of this study has been deposited in the figshare repository (https://doi.org/10.6084/m9.figshare.5248489.v1). All other data are available from the corresponding authors upon reasonable request.

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

## Acknowledgements

We gratefully acknowledge Dr. T.P. Ying and Dr. J.G. Cheng for measuring physical properties and valuable discussions. This work was supported by the National Natural Science Foundation of China (No. 51532010), the National Key Research and Development Program of China (Project No. 2016YFA0300600, 2016YFA0300504), the Starting-up for 100 talent of Chinese Academy of Sciences, the National Natural Science Foundation of China (91422303, 51522212, 11574394, 11774423), and the Fundamental Research Funds for the Central Universities, and the Research Funds of Renmin University of China (RUC) (15XNLF06, 15XNLQ07).

## Author contributions

J.G.G. and X.L.C. provided strategy and advice for the material exploration. J.G.G. and X.C. performed the sample fabrication, measurements and fundamental data analysis. N.L. carried out the theoretical calculation. X.J., H.C.L. and S.Y.L. measured the low

temperature properties and IV curves. Q.H.Z. and L.G. measured the HAADF images and advised for the structure analysis. J.G.G. and X.L.C. wrote the manuscript based on discussion with all the authors.

## Additional information

**Competing interests:** The authors declare no competing financial interests.

