## [Peer Review File · Nature Communications]

Reviewers' comments:

Reviewer #1 (Remarks to the Author):

This paper reports on the discovery of novel superconducting material $\text{AuTe}_2\text{Se}_4/3$. Using the detailed analysis of HAADF images, the authors successfully determined the crystal structure of $\text{AuTe}_2\text{Se}_4/3$. Very interestingly, the structure consists of molecule-like $\text{Au}_6\text{Se}_8\text{Te}_{12}$ cubes, which are connected via anisotropic Te-Te covalent bonds that result in a unique layered structure. Moreover, the authors observed superconductivity at $T_c = 2.75$ K. In addition, they observed the occurrence of the Berezinsky-Kosterlitz-Thouless topological transition, which is a hallmark of the two-dimensional nature of superconductivity.

I believe that the findings are interesting and attract a broad notice in the field of solid-state physics as well as chemistry. Thus, I recommend that the paper should be accepted for publication in Nature Communications after minor revisions.

I would like to ask authors to consider the following points:

- (1) Crystallographic data, which include crystal system, space group, lattice parameters, Z value, and fractional atomic coordinates of Au, Te, and Se, should be summarized in Tables. I understand that the authors have failed in determining the structural parameters using X-ray diffraction data, as described in the main text. But, I still would like to ask authors to present their crystallographic data that were determined by their HAADF images or by other means. Such data are also required to support their DFT calculations.
- (2) Powder x-ray diffraction data in Fig. S1 should be compared with data that are calculated using the aforementioned structural parameters.
- (3) There are many typo in Reference list: p.5905 should be p.6905 in Ref. 20; vol.113 should be vol.134 in Ref.21; and so on.

Reviewer #2 (Remarks to the Author):

The authors of this manuscript report a novel two-dimensional (2D) superconductivity occurred in a bulk sample of $\text{AuTe}_2\text{Se}_4/3$ by introducing a more electronegative Se atoms as compared with Te into the non-superconducting AuTe_2 lattice. They have performed studies and analysis on the sample in terms of its 3-dimensional structure, electronic band structure and the BKT transition characteristics. They demonstrated that the observed 2D superconductivity is likely originated from a quasi-2D layer structure in $\text{AuTe}_2\text{Se}_4/3$ leading to the nesting of quasi-1D bands, which enjoy a strong electron-phonon coupling. This novel discovery should be of high interest to a large community of researchers in condensed matter physics and thus the manuscript is suitable for publication in Nature Communication. However, the current version of the manuscript contains a number of minor issues and careless mistakes in presenting the figures, which should be revised/corrected before its publication should be considered. Below list these areas needed for revision and improvement:

A. Minor issues

- 1) Line 150-152: the authors claimed that there are almost 10% excess Au atoms that possibly occupy the interstitial sites of the cubic framework of $\text{AuTe}_2\text{Se}_4/4$. This claim likely comes from the average EDS results presented in Line 94. It is well known that the accuracy of EDS can be affected by various factors and usually up to a few % of uncertainty in elemental composition is expected. If there were 10% excess Au atoms incorporated as interstitials in the $\text{AuTe}_2\text{Se}_4/3$ lattice, the authors should be able to detect them in their presented high-resolution TEM images. Thus this claim seems to be groundless. Considering the fact that the authors were not able to determine the crystal structure from the powder x-ray diffraction pattern displayed in Fig. S1, a possible cause of the "excess" Au atoms may come from other structural phases in the sample.

Unless the authors could provide further experimental evidence in support of the claim of interstitial incorporation of excess Au, it should be removed or revised.

2) Line 202 – 207: the current presentation regarding the temperature dependence of the upper critical field and the deduction of the coherence length is too brief. In fact, the authors have not addressed the linear $(1 - T/T_c)$ dependence for the perpendicular critical field, and the square-root of $(1 - T/T_c)$ dependence for the parallel critical field as expected from the GL theory for 2D superconductors.

B. Errors in some figures

- 1) Fig. 5 uses "a" instead of "delta" for the exponent of the power law, inconsistent with the text.
- 2) Line 223 -225: The sentence "We can see that below T_c , it grows rapidly as temperature approaching T_c ", is a bit confusing, perhaps it should be rewritten as "it grows rapidly as temperature approaching T_c from the high-temperature side".
- 3) Fig. 1(a) and (b) are inconsistent with the corresponding figure caption and the description in the text.
- 4) In the "Big" Fig. 1 (page 22), the zone axis indices in (c) to (f) are inconsistent with the description in the text.

In addition, the current version of the manuscript contains quite a few typo or grammatical errors, which should be fixed. It is suggested that the authors get editing help from someone with full professional proficiency in English.

Reviewer #3 (Remarks to the Author):

In the present manuscript entitled "2D Superconductivity from Dimerization of Atomically Ordered AuTe₂Se_{4/3} Cubes", Guo and colleagues first discover the superconductivity in bulk AuTe₂Se_{4/3} Cubes. Furthermore, from the transport measurement, they discuss the 2D superconducting behavior of this system in terms of GL model and BKT transition. This result is interesting in the growing field of 2D superconductors.

The manuscript is well written, and the experimental results are convincing. However, BKT analysis seems to be wrong and the evidence for 2D superconductivity is not enough, which affect the main conclusion of this paper. Thus, I feel that there are some important points that need to be clarified before this paper is considered for publication in Nature Communications.

1) At page 1, second and third paragraph seems not to be related the main claims in the present paper. Authors should reduce this part, especially the introduction of materials. Instead, authors may want to cite the review papers about recently emerged 2D superconductors, e.g., Nature Reviews Materials, 2, 16094 (2016).

2) At page 7, line 197, the authors say "The resistivity suddenly drops to zero due to superconducting transition at T_c onset = 2.85 K". This sudden drop of resistive transition seems strange. In general, 2D superconductors show the broadening of resistive transition because of the superconducting fluctuation. The fluctuation behavior can be discussed in terms of Aslamazov-Larkin and Maki-Thompson term. Authors should discuss these terms For a more detailed treatment, check the following two papers.
Glatz et al., Phys. Rev. B 84 104510 (2011)
Baturina EPL 97, 17012 (2012).

3) At page 7, line 203, the authors say "The upper critical field $\mu_0 H_{c2}(0)$, as shown in Fig. 4(d), can be determined by Ginzburg-Landau theory at 50% of the normal resistivity at various fields".

In Fig. 4d, both in-plane and out-of-plane H_{c2} seem to show square-root behavior. In 2D GL model, in-plane H_{c2} show square-root behavior, while out-of-plane H_{c2} show linear behavior.

4) At page 9 (2D BKT transition), the way to determine BKT transition temperature is wrong. Fittings should be done at the zero current limit. In the present manuscript, however, the authors fit the data at the high excitation current, where the vortices and antivortices pairs are easily unbinding. Authors should show the wider range of I-V curve down to the lowest current/voltage limit. Check the following paper.

A.F. Hebard et al. Phys. Rev. Lett. 50, 1603 (1983).

4) Actually, the BKT transition is not enough evidence for 2D superconductors. Authors should show the angular dependence of H_{c2} . If it is 2D superconductors, the data should be well fitted by 2D Tinkham model. See also Saito et al. Science 350, 409 (2015).

5) Reference 43 is not appropriate. Instead, authors should cite Cao et al. Nano Letters 15, 4914 (2015).

6) At page 10, references 46 and 47 are not appropriate. Here, authors comment on the change of T_c and H_{c2} in Rashba-type systems. In that context, authors should cite famous Rashba-system, e.g., Nature 456,724 (2008), PRL 111, 057005 (2013).

Response to Referees

Reply to the Referee 1

1. **Comment:** *Crystallographic data, which include crystal system, space group, lattice parameters, Z value, and fractional atomic coordinates of Au, Te, and Se, should be summarized in Tables. I understand that the authors have failed in determining the structural parameters using X-ray diffraction data, as described in the main text. But, I still would like to ask authors to present their crystallographic data that were determined by their HAADF images or by other means. Such data are also required to support their DFT calculations.*

Reply: we agree with the referee and presented the crystallographic data as Table S1 in the supplementary materials. This is also the structural information we used in the DFT calculations.

2. **Comment:** *Powder x-ray diffraction data in Fig. S1 should be compared with data that are calculated using the aforementioned structural parameters.*

Reply: We calculated XRD pattern as the structural information and added it as the lower panel of Fig. S1. The experimental pattern shows that the 00l peaks are very strong in intensity and only a few weak peaks due to other diffractions are present, suggesting a preferred orientation feature, which agree well with the calculated one in peak position but not in intensity. Therefore, we use the HAADF-images to directly determine the crystal structure.

3. **Comment:** *There are many typo in Reference list: p.5905 should be p.6905 in Ref. 20; vol.113 should be vol.134 in Ref.21; and so on*

Reply: We apologize for these typos in reference list. We carefully went through the whole manuscript and reference list, and corrected the typos and syntaxes.

Reply to the Referee 2

1. **Comment:** *Line 150-152: the authors claimed that there are almost 10% excess Au atoms that possibly occupy the interstitial sites of the cubic framework of AuTe₂Se₄/4. This claim likely comes from the average EDS results presented in Line 94. It is well known that the accuracy of EDS can be affected by various factors and usually up to a few % of uncertainty in elemental composition is expected. If there were 10% excess Au atoms incorporated as interstitials in the AuTe₂Se₄/3 lattice, the authors should be able to detect them in their presented high-resolution TEM images. Thus this claim seems to be groundless. Considering the fact that the authors were not able to determine the crystal structure from the power x-ray diffraction pattern displayed in Fig. S1, a possible cause of*

the “excess” Au atoms may come from other structural phases in the sample. Unless the authors could provide further experimental evidence in support of the claim of interstitial incorporation of excess Au, it should be removed or revised.

Reply: We agree with the referee in excess Au. Initially we cannot understand the discrepancy between experimental and calculated atomic ratios, and the referee’s suggestions are more reasonable to explain the discrepancy. So we revised the related sentences in the page 5.

2. Comment: *Line 202 – 207: the current presentation regarding the temperature dependence of the upper critical field and the deduction of the coherence length is too brief. In fact, the authors have not addressed the linear $(1 - T/T_c)$ dependence for the perpendicular critical field, and the square-root of $(1 - T/T_c)$ dependence for the parallel critical field as expected from the GL theory for 2D superconductors.*

Reply: Thanks for pointing out this detail. We re-fitted the temperature-dependence of critical field by different Ginzburg-Landau equations in the context. The results were shown in the Fig. S5 (a), and a brief discussion part were added in the page 7.

3. Comment: *Errors in some figures*

1) *Fig. 5 uses “a” instead of “delta” for the exponent of the power law, inconsistent with the text.*

2) *Line 223 -225: The sentence “We can see that below T_c , …… it grows rapidly as temperature approaching T_c ”, is a bit confusing, perhaps it should be rewritten as “it grows rapidly as temperature approaching T_c from the high-temperature side”.*

3) *Fig. 1(a) and (b) are inconsistent with the corresponding figure caption and the description in the text.*

4) *In the “Big” Fig. 1 (page 22), the zone axis indices in (c) to (f) are inconsistent with the description in the text.*

Reply:

1) The exponent of power law has been changed into η for avoiding confusion.

2) We revised the sentence as per the referee’s comments.

3) We removed the wrong description and rewrote the captions.

4) The correct zone axis indices have been modified.

About the typos and syntaxes, we checked the whole manuscript and captions again, and carefully fixed above grammatical errors and typos. Also we got the help from professor to improve the English quality of manuscript.

Reply to the Referee 3

1. Comment: *At page 1, second and third paragraph seems not to be related the main claims in the*

present paper. Authors should reduce this part, especially the introduction of materials. Instead, authors may want to cite the review papers about recently emerged 2D superconductors, e.g., Nature Reviews Materials, 2, 16094 (2016).

Reply: We rewrote the second paragraph of introduction by deleting the contents with less irrelevance. The more appropriate examples of 2D materials have been added and the recommended literature has been cited.

2. Comment: *At page 7, line 197, the authors say “The resistivity suddenly drops to zero due to superconducting transition at T_c onset = 2.85 K”. This sudden drop of resistive transition seems strange. In general, 2D superconductors show the broadening of resistive transition because of the superconducting fluctuation. The fluctuation behavior can be discussed in terms of Aslamazov-Larkin and Maki-Thompson term. Authors should discuss these terms For a more detailed treatment, check the following two papers. Glatz et al., Phys. Rev. B 84 104510 (2011) Baturina EPL 97, 17012 (2012).*

Reply: We analyzed the resistivity drop according to Aslamazov-Larkin and Maki-Thompson effects of superconducting fluctuation following referee’s comments. The fluctuation effects were found to be present and the 2D order parameters develop at $T_{c0}=2.81$ K. The related details have been summarized in supplementary materials and Fig. S4, and the corresponding discussion has been added in page 7. The recommended literatures also have been cited.

3. Comment: *At page 7, line 203, the authors say “The upper critical field $\mu_0 H_{c2}(0)$, as shown in Fig. 4(d), can be determined by Ginzburg-Landau theory at 50% of the normal resistivity at various fields”. In Fig. 4d, both in-plane and out-of-plane H_{c2} seem to show square-root behavior. In 2D GL model, in-plane H_{c2} show square-root behavior, while out-of-plane H_{c2} show linear behavior.*

Reply: Thanks referee for pointing out this. We fitted the obtained data H_{c2} based on two Ginzburg-Landau equations and the fitted results are shown in the Fig. S5 (a). The related coherence length and superconducting thickness have been obtained. A brief discussion has been added in the page 7.

4. Comment: *At page 9 (2D BKT transition), the way to determine BKT transition temperature is wrong. Fittings should be done at the zero current limit. In the present manuscript, however, the authors fit the data at the high excitation current, where the vortices and antivortices pairs are easily unbinding. Authors should show the wider range of I-V curve down to the lowest current/voltage limit. Check the following paper: A.F. Hebard et al. Phys. Rev. Lett. 50, 1603 (1983).*

Reply: We appreciate the referee’s critical comments. The accurate BTK temperature indeed should be determined in the low current limit. We re-determined the BKT transition temperature again by using the data in the lowest current/voltage range. Some noises at $T= 1.8$ K and 2.2 K have been removed for clarity. We plotted the IV curves and temperature dependent power law exponent (η) in Fig. 5 (a) and (b). The new transition temperature is close to 2.8 K as shown in Fig. 5(b). The corresponding text has been added in page 8. Also, the recommended literature has been cited.

5. Comment: *Actually, the BKT transition is not enough evidence for 2D superconductors. Authors should show the angular dependence of H_{c2} . If it is 2D superconductors, the data should be well fitted by 2D Tinkham model. See also Saito et al. Science 350, 409 (2015).*

Reply: Following the referee's suggestion, we measured the angular dependence of resistivity and fitted the data using the 2D Tinkham and 3D model, respectively. It was found that the data match the 2D model well, which further confirmed the nature of 2D superconductivity in $\text{AuTe}_2\text{Se}_{4/3}$. The corresponding figures have been added as Fig. 4(d) and 4(e), and a brief discussion has been added in the page 7. The recommended literature has been cited as well.

6. Comment: *Reference 43 is not appropriate. Instead, authors should cite Cao et al. Nano Letters 15, 4914 (2015).*

Reply: Reference 43 has been replaced by the Cao's paper.

7. Comment: *At page 10, references 46 and 47 are not appropriate. Here, authors comment on the change of T_c and H_{c2} in Rashba-type systems. In that context, authors should cite famous Rashba-system, e.g., Nature 456,724 (2008), PRL 111, 057005 (2013).*

Reply: Thanks for recommending the literatures, and both papers have been cited in the context.

Reviewers' comments:

Reviewer #1 (Remarks to the Author):

I would like to thank authors for providing crystallographic parameters in Table S1. The authors compared the observed powder x-ray diffraction (PXRD) data with a profile that was calculated using the parameters, as shown in Fig. S1. However, I find severe disagreements between the observed data and the authors' simulation. In addition, I find that the authors' simulation is not consistent with a simulation which I have made using the provided crystallographic parameters.

First of all, I find that the simulated PXRD pattern in Fig. S1 is unusual in that the intensity is almost invisible at higher diffraction angles. I have performed a simulation using the provided crystallographic parameters and depicted the result in the attached Fig. 1(a) together with the authors' simulation in Fig. 1(b). Disagreements at higher angles are evident. I do not understand the reason why the authors' simulation intensity at higher angles is so small.

Secondary, both the simulated PXRD patterns in Fig. 1(a) and 1(b) fail to explain the observed PXRD intensity. There exist significant disagreements, which are evident at high diffraction angles larger than 25 degrees, between the simulations and the observation. A preferred orientation, which was suggested by authors in response letter, cannot be a reason of these disagreements because the very weak intensity of 002, 004, 005, 006 and 006 peaks suggest no orientation. I suggest that the 003 peak in Fig. S1 was incorrectly assigned. It could be a 310 peak, and the 003 peak could appear at a slightly lower angle, as seen in Fig. 1(a).

The disagreements between the observation and simulations cast a severe question about the validity of the structural parameters in Table S1. I admit that the authors could not perform structural refinement using the observed PXRD patterns. However, simulations should reasonably explain the PXRD observations, otherwise structural parameters are invalid.

I suggest that the manuscript in the present form is not suitable for publication in Nature Communications.

Reviewer #2 (Remarks to the Author):

The revised manuscript and the supplementary information indicate that the authors have put effort in giving point-by-point responses to the comments offered by the reviewers. However, some of the newly reported results raised a big concern regarding the claim that the observed 2D superconductivity comes from a confinement in the ab-plane of the layered structure of AuTe₂Se_{4/3}, which is realized by anisotropic stacking of cubes with diverse Te-Te covalent bonds and the consequent reduction in dimensionality. Below lists some unconvincing or contradictory issues that can be found in the revised manuscript and supplementary information:

1) Fig. 4 (e) is not convincing to show the 2D nature of the observed superconductivity as if the expected curves for 2D and 3D models are so similar, within the uncertainty of the measured data, the data fitting could go either way.

2) The authors follow the same suggestion offered by two reviewers regarding H_{c2}(T) vs T and carried out relevant fitting and reported the fitted coherence length at 0K to be 28.7 nm and the superconducting thickness of the sample to be 10.8 nm. However, these results are inconsistent with the claim that the quasi-2D unit cell of AuTe₂Se_{4/3} has a thickness of about 1nm as shown in Fig. 2 (g), no matter one considers the fitted data are referred to a single superconducting 2D unit or the accumulated results of the whole bulk sample with a thickness of 20-50 microns. (Remark: in the "Method" section, the authors mentioned the use of the focused ion beam approach to cut

narrow samples (thickness: ~50 nm), I guess here "thickness" actually refers to the "width" of the narrow device).

3) The newly added simulated PXRD spectrum together with the experimental PXRD spectrum as shown in Fig. S1 indicate that the sample is dominated by other unknown phases rather than the layered structure as presented by the reported TEM images. This raises a question why it is so easy to get the quasi-2D structure for performing the TEM imaging if the whole bulky sample only contains a very small amount of such a phase.

4) As the authors mentioned that the TEM samples were obtained through exfoliating the crystal by standard scotch tape method, then perhaps the reported layer structure of $\text{AuTe}_2\text{Se}_4/3$ may only exist at the surface of the as-grown crystal, which may explain the 2D nature of the superconductivity as observed by the transport and magnetic characterizations, however, if this is the case, the novelty of the current work will become much less important.

On top of the above, the English of the manuscript has not been polished enough, as one can easily find a few grammatical errors even just in the Abstract, such as "reduction (in) dimensionality", "directly exhibit (that)" and "cleavable nature (of) $\text{AuTe}_2\text{Se}_4/3$ ".

Taking all the above issues into account, I find it difficult to recommend the manuscript for publication in Nature Communications.

Reviewer #3 (Remarks to the Author):

The authors have done an excellent job in responding to all of the referees' comments and criticisms. Especially, the analysis for 2D superconductors including BKT transition and Tinkham model was improved. Now, I recommend the publication in Nature Communications.

Response to Referees

Reply to the Referee 1

1. Comment: *First of all, I find that the simulated PXRD pattern in Fig. S1 is unusual in that the intensity is almost invisible at higher diffraction angles. I have performed a simulation using the provided crystallographic parameters and depicted the result in the attached Fig. 1(a) together with the authors' simulation in Fig. 1(b). Disagreements at higher angles are evident. I do not understand the reason why the authors' simulation intensity at higher angles is so small.*

Secondary, both the simulated PXRD patterns in Fig. 1(a) and 1(b) fail to explain the observed PXRD intensity. There exist significant disagreements, which are evident at high diffraction angles larger than 25 degrees, between the simulations and the observation. A preferred orientation, which was suggested by authors in response letter, cannot be a reason of these disagreements because the very weak intensity of 002, 004, 005, 006 and 006 peaks suggest no orientation. I suggest that the 003 peak in Fig. S1 was incorrectly assigned. It could be a 310 peak, and the 003 peak could appear at a slightly lower angle, as seen in Fig. 1(a).

The disagreements between the observation and simulations cast a severe question about the validity of the structural parameters in Table S1. I admit that the authors could not perform structural refinement using the observed PXRD patterns. However, simulations should reasonably explain the PXRD observations, otherwise structural parameters are invalid.

Reply: Thanks referee for pointing out this mistake. Since the experimental PXRD pattern is not good enough for resolving crystal structure, we have to use the HAADF images to directly determine the atomic sites and the lattice periodicity. In so doing, we tried a lot of crystallographic sites for Au, Te and Se atoms and produced a few possible solutions for the crystal structure. The correct structure is among them, which was chosen based on other evidence. After carefully checking the simulated pattern in Supplementary Information and referee's comments, we found that the simulated pattern actually came from an early model of crystal structure, it does not correspond to the final crystallographic information (CIF file) we presented. We apologized for this carelessness.

After realizing this serious issue, we immediately simulated a new PXRD pattern using the final crystallographic information and found that this pattern can agree with the referee's simulation very well. The intensity of higher angle peaks now increases as expected. Meanwhile, we also agree with the referee's peak assignments about 310 and 003 peaks, so there is no preferred orientation in experimental PXRD pattern. So we removed the corresponding texts in the page 3 of main text.

In addition, we also carefully indexed the experimental and simulated PXRD patterns. According to the yielded peak indices and referee's suggestions, we can successfully index all the high intensity

peaks in low angle range. All the indices of high intensity peaks in experimental PXRD can match the simulated ones very well except minor impurity peaks. The weak peaks from minor impurity are labeled as red stars (*), and the two broad and weak peaks at high angle, labeled as symbol (+), cannot be accurately assigned due to multiple possibilities. Based on the consistency of both patterns, we now believe that the crystal structure is valid with a satisfactory accuracy of atomic sites determined by electron microscopy. The new simulated pattern and peak indices are shown in Fig. 1S (b).

We hope that above explanations could relieve referee's concern about the crystal structure.

 Reply to the Referee 2

1. Comment: Fig. 4 (e) is not convincing to show the 2D nature of the observed superconductivity as if the expected curves for 2D and 3D models are so similar, within the uncertainty of the measured data, the data fitting could go either way.

Reply: According to the 2D Tinkham and 3D anisotropic mass formulas, both simulated curves are indeed quite close as the angle is far way from 90°. However, they show difference in curvature as the angle approaches to 90°, in which the 2D curve is more sensitive to the angle than that of 3D curve.

Figure R1: The angle dependent the upper critical field H_{c2} for 2D superconductivity in ion-gated MoS_2 (Left figure, Figure 2d in Nat. Phys. **12**, 144, 2016) and ion-gated ZrNCl (Right figure, Figure 2c in Science **350**, 459, 2015). It can be seen that 2D and 3D curves can only be distinguished as the angle is near 90°.

The 2D superconductivity in ion-gated MoS_2 and ZrNCl are taken here as examples to illustrate this issue. Indeed, the H_{c2} is too close to be distinguished by 2D and 3D model as the angle is far way from 90° as shown in Fig. R1. However, one can find 2D curve is more sensitive to as the angle is near 90° and determine the superconducting dimensionality from the data near 90°. In the revised

manuscript, we magnified the portion of 2D and 3D curves near 90° and put it in the inset of Fig. 4(e) for clarity. We can conclude that the 2D model can explain the data of $\text{AuTe}_2\text{Se}_{4/3}$ very well, rather than a 3D model.

2. Comment: *The authors follow the same suggestion offered by two reviewers regarding $Hc_2(T)$ vs T and carried out relevant fitting and reported the fitted coherence length at 0K to be 28.7 nm and the superconducting thickness of the sample to be 10.8 nm. However, these results are inconsistent with the claim that the quasi-2D unit cell of $\text{AuTe}_2\text{Se}_{4/3}$ has a thickness of about 1nm as shown in Fig. 2 (g), no matter one considers the fitted data are referred to a single superconducting 2D unit or the accumulated results of the whole bulk sample with a thickness of 20-50 microns. (Remark: in the “Method ” section, the authors mentioned the use of the focused ion beam approach to cut narrow samples (thickness: ~50 nm), I guess here “thickness ” actually refers to the “width ” of the narrow device).*

Reply: The narrow sample with a width of 50 nm and thickness of 20~50 microns mentioned in the Method is only used for TEM observations. We cut the very narrow sample in order to observe the atomic stacking along c -axis. Otherwise, if the sample is too thick, we cannot obtain the crystallographic information and completely determine the crystal structure. In the last manuscript, we did not give out a precise description, which is misleading. So we revised the corresponding text in the Method.

The 2D thickness of superconductivity (d_{sc}) seems not be confined into the length of unit cell based on the reports so far. For instance, the d_{sc} in ion-gated ZrNCl is 1.8nm, which is less than one unit cell of ZrNCl (Science **350**, 459, 2015). However, the d_{sc} of ion-gated MoS_2 is 1.5nm, which is around the thickness of 3 unit cells (3×0.65 nm) and exceed the limitation of one unit cell (Nat. Phys. **12**, 144, 2016). In addition, 2D superconductivity thickness in thin flake of MoC_2 (Nat. Mater. **14**, 1135, 2015) is estimated to be 4 nm, which equals to eight unit cell of MoC_2 (8×0.5 nm).

The 2D superconductivity thickness is possible to be overestimated even with the state-of-art technique. One possibility is that the Cooper pairs could diffuse into the neighboring non-superconducting layer or impurity. Another possibility is that the spin-alignment destruction for Cooper pairs is comparable to the magnitude of coupling between magnetic field and the electron moment. The other source might come from the misalignment of the magnetic field or impurity scattering in sample, in which the magnetic field may not be precisely parallel or vertical to the surface of sample. In present data, we cannot rule out a possible overestimation of d_{sc} due to these sources.

3. Comment: *The newly added simulated PXRD spectrum together with the experimental PXRD spectrum as shown in Fig. S1 indicate that the sample is dominated by other unknown phases rather*

than the layered structure as presented by the reported TEM images. This raises a question why it is so easy to get the quasi-2D structure for performing the TEM imaging if the whole bulky sample only contains a very small amount of such a phase.

4) As the authors mentioned that the TEM samples were obtained through exfoliating the crystal by standard scotch tape method, then perhaps the reported layer structure of AuTe₂Se_{4/3} may only exist at the surface of the as-grown crystal, which may explain the 2D nature of the superconductivity as observed by the transport and magnetic characterizations, however, if this is the case, the novelty of the current work will become much less important.

Reply: The weak peaks at 29°, 44°, 48° and 76° in Fig. S1(a), labeled as red star, come from the powder sample with minor impurity. The mass ratio is empirically estimated below 10% as the peak-area ratios in experimental PXRD pattern. The property characterizations are actually performed on single crystals that are free of impurity.

As the reply to referee 1, we used the early crystal structure and presented a wrong simulation pattern in last manuscript, which possibly cause misunderstandings. This time, I put the correct simulated PXRD pattern deriving from the final crystal structure into Fig. S1 (b), it can be seen that almost all the high intensity peaks can match the simulated PXRD except minor impurities peaks. So the AuTe₂Se_{4/3} phase is the main phase, and we can easily separate the single crystals from sintered ingots.

We exfoliated the samples many times and measured the HAADF images of samples with varying thickness (50-200nm). All HAADF images show identical atomic distribution in the *ab*-plane. It can demonstrate that the layer structure of AuTe₂Se_{4/3} persists throughout the whole crystal. As for the surface superconductivity issue, we think the magnetic susceptibility of single crystal, i. e. ~90% superconducting volume fraction estimated from large diamagnetic signal at 1.8 K, could demonstrate that the observed superconductivity spreads through whole sample, rather than the surface superconductivity.

We hope that above explanation could relieve referee's concern about the sample issue. In addition, we have polished the English of the manuscript again and removed some typos and syntaxes in the text.

Reply to the Referee 3

We would like to thank referee 3 for improving the quality of whole manuscript.

REVIEWERS' COMMENTS:

Reviewer #1 (Remarks to the Author):

The authors have clarified all comments and criticisms that had made by referees. Especially, the sample issue was clarified in this revision. Now, I recommend the publication in Nature Communications.

Reviewer #2 (Remarks to the Author):

Through their further detailed responses, the authors have clarified the queries and puzzles raised by the reviewers. As the coherence length perpendicular to the layers is much larger than the unit-cell thickness in this layered material, "Quasi-2D superconductivity" should be used in the manuscript. With this change, the present form of the manuscript is suitable for publication in Nature Communications.

Response to Referees

Reply to the Referee 1

1. Comment: *The authors have clarified all comments and criticisms that had made by referees. Especially, the sample issue was clarified in this revision. Now, I recommend the publication in Nature Communications.*

Reply: We want to appreciate the referee for improving our manuscript and recommendation.

Reply to the Referee 2

1. Comment: *Through their further detailed responses, the authors have clarified the queries and puzzles raised by the reviewers. As the coherence length perpendicular to the layers is much larger than the unit-cell thickness in this layered material, "Quasi-2D superconductivity" should be used in the manuscript. With this change, the present form of the manuscript is suitable for publication in Nature Communications.*

Reply: We agree with the referee's suggestions. The quasi-2D superconductivity has been used in our main claim part. We also want to appreciate the referee for improving our manuscript and recommendation.